# On the dice loss variants and sub-patching

**Hoel Kervadec**[1]                                                                                HOEL@KERVADEC.SCIENCE
[1] *Erasmus MC, Rotterdam, The Netherlands*

**Marleen de Bruijne**[1,2]                                                          MARLEEN.DEBRUIJNE@ERASMUSMC.NL
[2] *University of Copenhagen, Denmark*

## Abstract

The soft-Dice loss is a very popular loss for image semantic segmentation in the medical field, and is often combined with the cross-entropy loss. It has recently been shown that the gradient of the dice loss is a "negative" of the ground truth, and its supervision can be trivially mimicked by multiplying the predicted probabilities with a pre-computed "gradient-map" (Kervadec and de Bruijne, 2023). In this short paper, we study the properties of the dice loss, and two of its variants (Milletari et al., 2016; Sudre et al., 2017) when sub-patching is required, and no foreground is present. As theory and experiments show, this introduce divisions by zero which are difficult to handle gracefully while maintaining good performances. On the contrary, the mime loss of (Kervadec and de Bruijne, 2023) proved to be far more suited for sub-patching and handling of empty patches.

**Keywords:** Semantic segmentation, full supervision, dice loss

## 1. Background

The Dice coefficient, measuring overlap between two areas can be written as $\mathrm{DSC}(y, s; k) := \frac{2\left|\Omega_y^{(k)} \cap \Omega_s^{(k)}\right|}{\left|\Omega_y^{(k)}\right| + \left|\Omega_s^{(k)}\right|} = \frac{2\sum_{i \in \Omega} y^{(i,k)} s^{(i,k)}}{\sum_{i \in \Omega}\left[y^{(i,k)} + s^{(i,k)}\right]}$, with $\Omega \subset \mathbb{R}^D$ a $D$-dimensional image space, $y^{(\cdot,\cdot)} : (\Omega \times \mathcal{K}) \rightarrow \{0,1\}$ a ground-truth as a binary function, and $s^{(\cdot,\cdot)} : (\Omega \times \mathcal{K}) \rightarrow \{0,1\}$ a predicted segmentation. $\mathcal{K} = \{0, 1, ..., K\}$ is the set of classes to segment, $0$ being the background class and $K$ the number of object classes. $\Omega_y^{(k)} := \{i \in \Omega | y^{(i,k)} = 1\} \subseteq \Omega$ denotes the subset of the image space where $y$ is of class $k$. With continuous probabilities $s_{\boldsymbol{\theta}}^{(\cdot,\cdot)} \in [0, 1]$ we can define a Dice loss:

$$\mathcal{L}_{\mathrm{DSC}}(y, s_{\boldsymbol{\theta}}) := \frac{1}{|\mathcal{K}|} \sum_{k \in \mathcal{K}} \left(1 - \frac{2\sum_{i \in \Omega} y^{(i,k)} s_{\boldsymbol{\theta}}^{(i,k)}}{\sum_{i \in \Omega}\left[y^{(i,k)} + s_{\boldsymbol{\theta}}^{(i,k)}\right]}\right). \tag{1}$$

It has been shown (Kervadec and de Bruijne, 2023) that its gradient wrt. the softmax takes the following form:

$$\frac{\partial \mathcal{L}_{\mathrm{DSC}}}{\partial s_{\boldsymbol{\theta}}^{(i,k)}} = \begin{cases} \frac{-2\left(U^{(k)} - I^{(k)}\right)}{\left(U^{(k)}\right)^2} & \text{if } y^{(i,k)} = 1, \\ \frac{2I^{(k)}}{\left(U^{(k)}\right)^2} & \text{otherwise,} \end{cases} \tag{2}$$

with $I^{(k)} = \sum_{i \in \Omega} y^{(i,k)} s_{\boldsymbol{\theta}}^{(i,k)}$ and $U^{(k)} = \sum_{i \in \Omega}\left[y^{(i,k)} + s_{\boldsymbol{\theta}}^{(i,k)}\right]$. This means that the gradient of the dice loss takes only two different values over the whole image, as a weighted negative of y.

Moreover, (Kervadec and de Bruijne, 2023) has shown that the supervision of the Dice loss can be mimicked with the following simple loss:

$$\mathcal{L}_{\text{Mime}}(y, s_{\boldsymbol{\theta}}) := \boldsymbol{\omega}_y^\top \boldsymbol{s_{\theta}}, \tag{3}$$

with $\boldsymbol{\omega}_y \in \mathbb{R}^{|\mathcal{K}||\Omega|}$ a flattened, pre-computed gradient map, and $\boldsymbol{s_\theta} \in [0,1]^{|\mathcal{K}||\Omega|}$ the flattened predicted probabilities. With $\boldsymbol{y} \in \{0,1\}^{|\mathcal{K}||\Omega|}$ the flattened ground truth, we can simply do: $\boldsymbol{\omega}_y = -\boldsymbol{y}a + (1 - \boldsymbol{y})b$ with $a, b > 0$. In this paper, we set $a$ and $b$ based on the class distribution over the whole dataset $\mathcal{D} = \{(x_n, y_n)\}_{n=1}^N$, i.e. $a^{(k)} = \frac{1}{|\mathcal{D}| \sum_{n \in \mathcal{D}} |\Omega_{y_n}^{(k)}|}$ and $b^{(k)} = \frac{1}{|\mathcal{D}| \sum_{n \in \mathcal{D}} \left[|\Omega| - |\Omega_{y_n}^{(k)}|\right]}$

Some well-known variants have been introduced to better handle imbalanced tasks. The Generalized Dice Loss (Sudre et al., 2017) is based on the Generalized Dice *Score* (Crum et al., 2006):

$$\mathcal{L}_{\text{GDL}}(y, s_{\boldsymbol{\theta}}) := 1 - \frac{2 \sum_{k \in \mathcal{K}} w^{(k)} \sum_{i \in \Omega} y^{(i,k)} s_{\boldsymbol{\theta}}^{(i,k)}}{\sum_{k \in \mathcal{K}} w^{(k)} \sum_{i \in \Omega} \left[y^{(i,k)} + s_{\boldsymbol{\theta}}^{(i,k)}\right]}, \tag{4}$$

with $w^{(k)} = \frac{1}{\left(\sum_{i \in \Omega} y^{(i,k)}\right)^2}$. V-Net (Milletari et al., 2016) slightly modify the base dice loss by squaring the denominator probabilities:

$$\mathcal{L}_{\text{VNet}}(y, s_{\boldsymbol{\theta}}) := \frac{1}{|\mathcal{K}|} \sum_{k \in \mathcal{K}} \left(1 - \frac{2 \sum_{i \in \Omega} y^{(i,k)} s_{\boldsymbol{\theta}}^{(i,k)}}{\sum_{i \in \Omega} \left[y^{(i,k)^2} + s_{\boldsymbol{\theta}}^{(i,k)^2}\right]}\right). \tag{5}$$

## 2. Sub-patching and empty patches

As the Dice overlap score is defined through the intersection and union of two areas, it cannot be "decomposed" in smaller computations: one cannot compute a series of Dice on subsets of $\Omega$, and then aggregate them to get the original dice score. This is an issue when training a neural network requires sub-patching—either a 3D sub-patch or a 2D slice—due to memory limitations. Computing the dice on the sub-patch is doable, but it loses its semantic meaning. More importantly, it increases the chance of encountering empty foregrounds ($\Omega_y^{(k)} = \Omega_s^{(k)} = \emptyset$) within the patch, which for all dice variants (1), (4) and (5) will cause divides-by-zero in various places. While it can be relatively mitigated through careful addition of small $\epsilon$ in their implementation, it is less than ideal and can introduce instabilities in the training.

## 3. Experiments

Experiments are performed with a lightweight 2D-ENet (Paszke et al., 2016) using the Adam optimizer (Kingma and Ba, 2014). We report the mean DSC and 95th percentile of the Hausdorff distance on the testing set for both datasets. For HD95, when no object is predicted, we count the diagonal of the scan. When there is no object to predict, and no object is predicted, we count 0. We evaluate on the following two datasets:

Table 1: Mean testing DSC (%) ↑ / HD95 (mm) ↓.

| Dataset / Loss | ACDC | | | | WMH | | |
|---|---|---|---|---|---|---|---|
| | RV | Myo | LV | **All** | WMH | Other pathologies | **All** |
| $\mathcal{L}_{\text{DSC}}$ | 77.2/11.8 | 79.2/04.9 | 90.3/03.2 | 82.2/06.7 | 68.6/009 | 00.5/251 | 34.6/130 |
| $\mathcal{L}_{\text{VNet}}$ | 78.0/13.7 | 78.4/03.8 | 89.9/05.8 | 82.1/07.8 | 70.4/007 | 00.2/251 | 35.3/129 |
| $\mathcal{L}_{\text{GDL}}$ | 78.3/14.6 | 80.4/03.8 | 90.2/06.0 | 83.0/08.1 | 08.2/088 | 00.0/286 | 04.1/187 |
| $\mathcal{L}_{\text{Mime}}$ | 81.5/09.7 | 80.2/03.4 | 90.9/04.0 | 84.2/05.7 | 61.1/006 | 63.0/135 | 62.1/071 |

$(a)$ GT     $(b)$ $\mathcal{L}_{\text{DSC}}$     $(c)$ $\mathcal{L}_{\text{VNet}}$     $(d)$ $\mathcal{L}_{\text{GDL}}$     $(e)$ $\mathcal{L}_{\text{Mime}}$

Figure 1: Example results from the WMH testing set.

**ACDC (Bernard et al., 2018)**  contains cine-MRI of the heart, providing annotations at systole and diastole of the right-ventricle (RV), myocardium (Myo) and left-ventricle (LV) so that $K = 3$. The dataset contains 100 patients with different pathologies. We kept 10 patients for validation and 20 for testing.

**WMH 1.0 (Kuijf et al., 2022)**  The full dataset of the *White Matter Hyperintensities (WMH)* MICCAI 2017 challenge contains annotations for the 60 scans of the training set (10 are kept here for validation) and 110 scans of the testing set. Additionally, the annotations also roughly segment *other pathologies* present in the scans, so that $K = 2$. This is a very imbalanced dataset, even more pronounced for the *other pathologies* class.

## 4. Results, discussion and conclusion

Metrics computed on the testing set are reported in Table 1 and Figure 1 shows a single slice of WMH testing set. We can see that all dice variants perform similarly on the ACDC dataset, which is to be expected. However, on WMH, all dice variants struggle with the "Other pathologies" class, while the hyperintensities are more-or-less well segmented. On the contrary, the mime loss proved able to handle more gracefully empty patches, which resulted in a better segmented "other pathologies" while maintaining performances on the main class.

To summarize, we discussed the limitations of the dice loss and some of its variants, with respect to sub-patching. Notably, all variants struggle when a patch is empty, as it introduce division by zero. On the contrary the Mime loss from (Kervadec and de Bruijne, 2023) can easily be sub-patched without introducing extra instabilities. Its simple definition also enables easy tuning with respect to the datasets imbalance.

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
