# OpenReview forum: "On the dice loss variants and sub-patching"
_MIDL.io/2023/Short_Paper_Track — MIDL 2023 Short paper track Poster_

### Official Review · Reviewer_9u1T · 2023-04-24

**Rating:** 8
**Confidence:** 4

**Review:**

# Summary

This paper seems to be based on a recent preprint which is currently under review, which analyses theoretical properties of the gradient of the Dice loss for segmentation. The authors propose an alternative to the Dice loss and compare it experimentally to existing variants of the Dice loss.



# Strengths

The authors' insights about the Dice loss are quite surprising - simple yet to my knowledge nobody has thought about this so far, and it will be interesting to ponder the consequences.

# Weaknesses

- As a comparison, I would appreciate it if the authors could also report results of training with a crossentropy loss.
- Would it be worthwile to also analyse the gradients of the different Dice variants, in particular the proposed "Mime" los
s?

---

### Official Review · Reviewer_CmGz · 2023-04-24
**A new dice loss variant for sub patches**

**Rating:** 7
**Confidence:** 4

**Review:**

This short paper study a variant of Dice loss referred to as mime loss introduced by the authors themselves (Kervadec and de Bruijne, 2023) and compare it with other variants of dice loss
(Milletari et al., 2016; Sudre et al., 2017). The goal is to achieve improved performance when dealing with subpatches of images (3D patches or 3D slices from a volume) and original Dice loss becomes less meaningful due to presence of empty patches and divisions by zero when using the Dice variants. The author’s mime loss handles these and shows improved performance and claimed to be better suited for sub-patching and handling of empty patches. The results are compared in two datasets. In the ACDC dataset, which includes right ventricle, myocardium and left ventricle segmentations in cine MRI of the heart, all variants of Dice loss performed similarly since subpatching did not result in empty patches. The second dataset was  white matter hyperintensities (WMH) MICCAI challenge data, which included an additional class indicating other pathologies, and was an imbalanced dataset. Therefore Dice variants had poor performance due to empty patches but mime loss had better performance. This is an interesting study and experimental results support the authors’ claims, however, there is no comparison to other (non-Dice) kinds of loss functions that deal with imbalanced data. Also, nowadays 3D data can fit in larger memories or multiple GPUs may be utilized to fit the data  in memory and subpatching may be suboptimal.